# A Reinforcement Learning-Based Congestion Control Approach for V2V Communication in VANET

**Xiaofeng Liu** †  , **Ben St. Amour** † **and Arunita Jaekel** *

School of Computer Science, University of Windsor, Windsor, ON N9B 3P4, Canada
* Correspondence: arunita@uwindsor.ca; Tel.: +1-519-253-3000 (ext. 2996)
† These authors contributed equally to this work.

**Abstract:** Vehicular ad hoc networks (VANETs) are crucial components of intelligent transportation systems (ITS) aimed at enhancing road safety and providing additional services to vehicles and their users. To achieve reliable delivery of periodic status information, referred to as basic safety messages (BSMs) and event-driven alerts, vehicles need to manage the conflicting requirements of situational awareness and congestion control in a dynamic environment. To address this challenge, this paper focuses on controlling the message transmission rate through a Markov decision process (MDP) and solves it using a novel reinforcement learning (RL) algorithm. The proposed RL approach selects the most suitable transmission rate based on the current channel conditions, resulting in a balanced performance in terms of packet delivery and channel congestion, as shown by simulation results for different traffic scenarios. Additionally, the proposed approach offers increased flexibility for adaptive congestion control through the design of an appropriate reward function.

**Keywords:** VANET; congestion control; reinforcement learning

## 1. Introduction

Road traffic safety is a persistent issue that has been the subject of study for over 80 years [1]. The safety of motor vehicles, as the most widely used mode of transportation, is of utmost importance. Despite the global pandemic in 2020, an estimated 38,680 people lost their lives in motor vehicle crashes, marking the highest projected number of fatalities since 2007 [2]. In addition to driver behavior and attitudes, improving vehicle safety through inter-vehicular communication is a crucial aspect. Vehicular ad hoc networks (VANETs) [3] have gained recognition from government agencies, automobile industries, and academia as key components of intelligent transportation systems (ITS) [4] to enhance safety and efficiency on the road. VANETs enable direct communication between vehicles through onboard units (OBUs) or with infrastructure nodes such as roadside units (RSUs), thereby facilitating the dissemination of safety-related information. This information, including a vehicle's position, speed, and acceleration, is periodically broadcast as basic safety messages (BSMs) [5] (also referred to as cooperative awareness messages (CAMs) [6] in Europe) to surrounding vehicles, enabling safety-critical applications, such as collision avoidance, lane change warnings, and hard braking warnings. The timely and accurate dissemination of these messages is essential for the effective operation of various safety applications.

The 5.9 GHz band has been allocated 75 MHz of spectrum for vehicle-to-vehicle (V2V) communication in VANET through dedicated short-range communications/wireless access for the vehicular environment (DSRC/WAVE) [7]. An additional 30 MHz of the spectrum is reserved for cellular vehicle-to-everything (C-V2X) communication [8]. Channel 172, with a bandwidth of 10 MHz, has been designated for vehicle safety and uses the 802.11p protocol [7], a contention-based random access MAC layer protocol. This type of communication can lead to simultaneous transmissions, causing packet collisions and reducing the reliability of communication [9]. As the number of vehicles increases, the

broadcasting of BSMs can easily lead to congestion on this single channel, resulting in lower reception probability and decreased transmission ranges. Traditional methods attempt to address this by controlling transmission parameters, such as the transmission power and transmission rate (also known as beacon rate), but this can also lead to reduced awareness and increased inter-packet delay (IPD) [10]. Although an optimal combination of transmission power and rate could potentially provide sufficient awareness while keeping the channel load below a specified threshold, the optimization problem is non-convex, making it challenging to solve, as noted in [11].

In vehicular ad hoc networks (VANETs), it is challenging to find optimal solutions for congestion and awareness due to the highly mobile nodes and dynamic environments. As an alternative, this paper approaches the problem as a decision-making task, where each vehicle must make the appropriate choice of transmission parameters for its safety messages based on the information it gathers from its surroundings. The transmission rate selection is formulated as a Markov Decision Process (MDP) [12], where each vehicle is modeled as an independent agent that interacts with its environment and selects its transmission parameters based on the current conditions. Reinforcement learning (RL) [13] is utilized to train the vehicles to make the right decisions, and the learning is based solely on the observations of the surrounding environment. After training, the vehicles will have learned to make optimal decisions in dynamic conditions, adapting to changes in vehicle density and channel load. The key contributions of this paper are:

- A framework to solve the MDP using RL methods, with a focus on discrete action and state spaces.
- A Q-learning algorithm, where the training data are obtained directly from a simulated dynamic traffic environment, allowing for a more realistic representation of state transitions by observing channel busy ratio (CBR) values for different transmission rates and vehicle densities.
- A reward function is defined, combining CBR and transmission rate, to keep the channel load under a target threshold while maximizing the transmission rate for congestion control.
- Our simulation results demonstrate that the proposed Q-learning approach is successful in maintaining the desired channel load under various dynamic traffic scenarios and exhibits a lower Beacon Error Rate (BER) compared to existing methods.

The structure of this paper is outlined as follows:

In Section 2, we conduct a review of current approaches for congestion control, with a specific emphasis on recent machine learning (ML)-based techniques. In Section 3, we present our formulation of congestion control as an MDP and introduce our proposed Q-learning-based algorithm for congestion control. In Section 4, we demonstrate the application of our framework with a concrete example and discuss the results obtained. Finally, in Section 5, we provide a conclusion and suggest directions for future work.

## 2. Related Work

VANETs are constantly evolving to support a diverse range of applications, including both critical safety applications, such as forward collision warnings and traffic signal violation warnings, and more comfort-oriented applications, such as weather information systems and restaurant recommendations [14]. These safety applications require the reliable delivery of both alert messages and the periodic broadcast of BSMs containing information, such as a vehicle's position, speed, and heading. However, with limited channel capacity and high transmission rates required for maintaining awareness, the reliable delivery of BSMs can pose a challenge in VANETs. In this section, we will explore both classical approaches and recently proposed approaches.

The nominal rate for transmitting BSMs is 10 packets per second. One well-known rate-based congestion control method is the linear message rate integration control (LIMERIC) algorithm [15]. Its objective is to fairly distribute the available channel bandwidth among all vehicles by dynamically adjusting the BSM transmission rate in each iteration. LIMERIC



uses linear feedback to adapt the transmission rate instead of the limit cycle behavior. In LIMERIC, vehicles in a given region, i.e., a single collision domain, where all the nodes measure the same channel load, sense and adapt their transmission rates to meet a specified CBR. LIMERIC is a distributed algorithm, executed independently by each station in a neighborhood. While LIMERIC effectively reduces congestion, it can result in an increased IPD from neighboring vehicles. The error model-based adaptive rate control (EMBARC) algorithm [16] improves LIMERIC with the capability to preemptively schedule messages based on the vehicle's movement, i.e., a kind of suspected tracking error technique. This approach can provide extra transmission opportunities for highly dynamic vehicles, which can therefore reduce instances of large tracking errors. Another rate-based approach, BSM rate control over IEEE802.11p vehicular networks (BRAEVE) [17], uses the estimated number of vehicles as the input parameter, leading to smoother convergence and lower packet error ratio, IPD, and tracking error compared to other algorithms. In a previous study [18], researchers introduce a congestion-aware message (CAM) for beacon signals in the vehicle environment that utilizes vehicle IDs. The CAM model incorporates the unique automobile IDs into the back-off procedure, weighting the randomized back-off numbers chosen by each vehicle. This results in the generation of car ID-based randomized back-off codes that reduce the risk of collisions caused by identical back-off numbers. In this paper, BSMs as the standard safety messages are not considered. The proposed scheme in [19] first analyzes congestion detection schemes and then utilizes a priority model to adjust the transmission rate of beacon messages. The Tabu search algorithm was employed to control network congestion. This paper does not show the performance in terms of CBR which is a critical metric to evaluate the channel load. In their paper [20], the authors present a solution to the message rate control problem by formulating it as a Markov decision process and using on-policy control with function approximation in reinforcement learning (RL). This approach enables vehicles to take appropriate actions quickly, with low computational cost, and converge in a short time. However, the action space consists of only three actions which severely limits the vehicle's ability to mitigate congestion.

In addition to the transmission rate, another adjustable parameter for congestion control is transmission power. By increasing the transmission power, a packet can reach a greater distance and potentially more vehicles, enabling control over the number of vehicles a BSM can reach. Transmission power is typically calculated based on the measured or estimated level of channel load. A widely cited power control algorithm is the distributed fair transmit power adjustment for VANETs (D-FPAV) [21]. D-FPAV achieves congestion control by setting the node transmission power based on the prediction of application-layer traffic and the observed number of vehicles in the surrounding area. As a proactive approach, D-FPAV uses a predefined maximum beaconing load (MBL) and calculates the network-wide optimal transmission power to keep the channel load under this value. By periodically gathering the information about AL (Application-layer Load) for nodes within the maximum carrier sense range, a node first uses FPAV [22] to calculate the maximum transmission power for all the nodes within its maximum carrier sense range, without violating the MBL condition. Then the nodes exchange this power by broadcasting to the neighbors within the maximum carrier sense range. After getting the same message from the nodes in its vicinity, each node computes the final power level, which is the minimum value among the local maximum transmission power levels in its vicinity. In FPAV, all the nodes increase their transmission power simultaneously, by the same amount, as long as the MBL condition is satisfied. Both D-FPAV and FPAV rely on predicting the MBL, which becomes increasingly challenging as vehicle density increases. The adaptive beacon transmission power (ABTP) algorithm [23] firstly improves the traditional linear prediction algorithm and elaborates a linear combination model to reduce the vehicle's turning error. The approach adjusts transmission power based on vehicle position prediction error, increasing power for vehicles with large errors and reducing it for vehicles with small errors. The paper lacks a mechanism for evaluating distance error, such as identifying



when the distance error is 10 m. Another limitation of the study is that it sets a maximum beacon load of 70%, which can be surpassed with high vehicle density.

The DSRC standard defines seven data rates, ranging from 3 Mbps to 27 Mbps, with 10 MHz channels, according to the reference in [7]. The goal of data rate control is to increase the rate by using higher modulation levels, which reduces channel load and collisions. Most studies in the literature rely on the findings of [24], which suggests that 6 Mbps is optimal for most cases. However, this study is not dynamic and the transmit power levels are not clear. The binary rate algorithm described in [25] is a data-rate control algorithm that dynamically adjusts the data rate based on the channel load. The algorithm utilizes a state machine with three states: relaxed, active, and restrictive. Each state has different transmission parameters, and the vehicle detects the current state based on the channel load and uses the corresponding parameters to send messages. It is worth noting that this algorithm can only switch between four data rates. In [26], the authors propose a new approach that dynamically selects an appropriate data rate for each BSM transmission, based on the current CBR. Unlike existing algorithms that typically increment the data rate only one level at a time, regardless of how high the CBR is, the proposed algorithm directly estimates the appropriate data rate to use based on the current CBR value. This allows the channel congestion to converge to the desired level much faster, leading to lower packet loss and improved packet delivery ratio. Similarly, when the current CBR is below the desired threshold, the proposed algorithm calculates the appropriate data rate and starts transmitting directly using these data rates, rather than moving through intermediate levels. This approach uses a fixed threshold of CBR, which could vary dynamically. Despite these efforts, the challenge of achieving higher signal-to-interference noise ratios (SINRs) and meeting safety range constraints for higher data rates remains. As a result, data rate control is always combined with other transmission parameters for congestion control in VANET.

The ultimate goal of congestion control is to enhance vehicle awareness. While decreasing the transmission power or rate may alleviate congestion, it reduces awareness quality. On the other hand, increasing awareness through transmission with maximum power and rate leads to increased congestion. As noted in [27], there is an inverse correlation between transmission power and rate at a constant target load on the wireless channel, meaning that decreasing the transmission power enables an increase in the transmission rate and vice versa. The existing approaches discussed, however, only allow for a single power/rate pair, resulting in a trade-off dilemma between transmission power and rate. Paper [28] proposes an RL Q-learning approach to balance the trade-off between transmission rate and power. The authors define a Markov decision process (MDP) by making a set of reasonable simplifying assumptions which are resolved using Q-learning techniques of RL. This RL-based approach decides the next state of the agent based on the possible actions and a Nakagami model instead of the interaction with the environment. In another paper [29], the authors present two mechanisms, transmission rate and data rate adaptation combined with power control mechanisms, called Combined power and message-rate adaptation (CPMRA) and combined power and data-rate adaptation (CPDRA). In CPMRA, the transmission power and transmission rate are adapted based on the current CBR value so that the congestion is maintained within specified limits. In CPDRA, The data rate and transmission power are adapted depending on the current congestion situation. The power control mechanism is the same as the CPMRA algorithm, and the ego node changes the transmit power based on available neighbors and CBR values. The definition of CBR in this paper is not the channel busy time ratio but the ratio of message length received and channel capacity which ignored the channel busy time caused by packet collision. In their paper [30], the authors introduce a congestion control algorithm called channel-aware congestion control (CACC) that can control both the transmission power and data rate. This algorithm takes into consideration the received signal strength (RSS) to diagnose packet loss and determine the channel conditions, which could be caused by either severe fading or channel congestion. If the channel experiences severe fading, the algorithm lowers the data rate to use a more

robust modulation and coding scheme that can better handle the noise. At the same time, it adjusts the transmission power to maintain an acceptable packet error rate. This approach only uses constant speed within a pre-determined mobility area and lacks techniques to estimate the channel state using packet loss more accurately in the broadcast environment.

In Table 1, we present a comparison of several notable congestion control approaches. The phrases "Tx. Rate" and "Tx. Power" refer to the transmission rate and power respectively, while "rule-based" indicates a traditional, non-machine learning-based approach and "learning-based" refers to approaches that utilize machine learning algorithms.

**Table 1.** Comparison of algorithms.

| Algorithm | Control Parameter | Rule-Based/ Learning-Based | Metrics Used | Message Type |
|---|---|---|---|---|
| LIMERIC [15] | Tx. Rate | Rule-based | CBR | Beacon |
| EMBARC [16] | Tx. Rate | Rule-based | Tracking error | Beacon |
| D-FPAV [21] | Tx. Power | Rule-based | Probability of message reception | Beacon, Event messages |
| SPAV [31] | Tx. Power | Rule-based | Beaconing load | Beacon |
| SBAPC [32] | Tx. Power | Rule-based | Beacon error rate, CBR, IPD | Beacon |
| DR-DCC [25] | Data Rate | Rule-based | PDR | Beacon |
| CACC [30] | Hybrid | Rule-based | PDR, PCR, CBR | Beacon |
| BH-MAC [33] | Hybrid | Rule-based | RAC | Beacon |
| CPMRA-CPDRA [29] | Hybrid | Rule-based | CBR | Beacon |
| ML-CC [34] | Tx. Rate | Learning-based | PLR, Average Delay | Beacon |
| MDPRP [28] | Hybrid | Learning-based | CBR, PDR | Beacon |
| SSFA [20] | Tx. Rate | Learning-based | CBR,PDR | Beacon |

*Discussion*

The complexity of congestion control in VANET, due to its dynamic nature, diversity, and mobility, has created significant challenges. As previously mentioned, adjusting transmission rate or power can have a negative impact on awareness, while a combination of both is an ideal solution but with an optimization problem that is not convex [11]. To tackle this challenge, researchers have proposed machine learning-based algorithms for congestion control that are based on real-time states instead of predetermined rules, allowing for better adaptability to the dynamic network environment. Supervised and unsupervised learning techniques have been used in the past, but these methods are limited in their ability to handle real-world network conditions as they are trained offline and cannot classify wireless and congestion loss accurately [35]. In contrast, RL techniques offer advantages such as online training capabilities.

This paper proposes a Q-learning-based congestion control algorithm for the field of VANET congestion and awareness control, demonstrating the application of RL-based methodology in this area. The proposed algorithm builds upon a previous publication [36] by providing a more comprehensive description of the RL framework and updating the Q-learning-based congestion control algorithm. A new estimation function has been introduced to accurately predict the CBR based on vehicle density and transmission rate, which was not present in the original publication. The paper also includes an expanded literature review and new simulation results for various performance metrics, including Packet Delivery Ratio (PDR) and BER.

## 3. An RL-Based Framework of Congestion and Awareness Control

In the field of VANET, congestion control is a vital challenge for ensuring the safety of communication over the limited bandwidth of the wireless channel. The objective of

congestion control is to alleviate channel congestion by reducing the number of transmitted safety messages. This can be achieved by increasing the time interval between transmissions. However, reducing the frequency of transmissions leads to a reduction in awareness, causing a decrease in the visibility of other vehicles and vice versa. This in turn can cause congestion to arise again when the density of vehicles increases. The vehicles must therefore balance between prioritizing congestion control and awareness control, which becomes particularly challenging in dynamic mobility environments. Making the correct decision in different scenarios is crucial for successful congestion control. For instance, when the vehicle density is low, it may be beneficial to increase the transmission power to achieve a larger transmission range, while still maintaining an acceptable channel load by adjusting the transmission rate. When the vehicle density is high, the transmission power and rate must be adjusted accordingly. This decision-making problem is influenced by various factors, including vehicle density, channel congestion, and packet delay, making it challenging to find an optimal combination of transmission parameters using traditional methods, especially since some constraints may conflict with each other. It is important to note that the decision-making process in VANET must take into consideration the current situation only. This is due to the Markov property, which states that the future is independent of the past given the present. This means that the vehicle's decision is based on the current state of the traffic flow and channel, rather than prior conditions. The Markov property can be formalized as:

$$P[S_{t+1}|S_t] = P[S_{t+1}|S_1, \ldots, S_t] \tag{1}$$

Equation (1) shows that the state at the next moment, $t + 1$, is solely dependent on the state of the current moment, $t$. Hence, we can model the problem with an MDP. RL is a good framework to find solutions to an MDP [13]. The main learning principle of a typical RL cycle in V2V communication can be described in Figure 1.

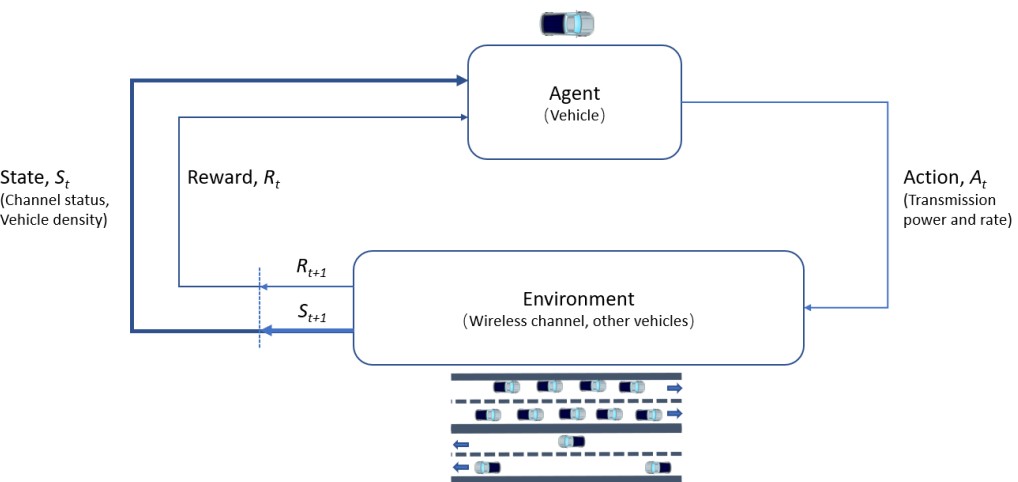

**Figure 1.** Typical RL Cycle in V2V communication.

Here, a vehicle and environment interact at each of a sequence of discrete time steps, $t = 0, 1, 2, 3, \ldots$. At each time step $t$, the vehicle receives some representation of the environment's state , $S_t \in \mathcal{S}$, and based on the representation, it selects an action, $A_t \in \mathcal{A}(s)$. After one time step, the vehicle will receive a numerical reward, $R_{t+1} \in \mathcal{R}$, and it moves to a new state, $S_{t+1}$. The vehicle's goal is to maximize the total reward it receives. This means maximizing not the immediate reward, but the cumulative reward in the long run. If the sequence of rewards received after time step $t$ is denoted as $R_{t+1}, R_{t+2}, R_{t+3}, \ldots$, the maximized reward, where the return is $G_t$, can be the sum of all the rewards at each time step until the final state. However, the BSM transmission is a continuous task without a time limit. To avoid the return being infinite, a discount rate $\gamma$ is used to determine the

present value of future rewards: a reward received $k$ time steps in the future is worth only $\gamma^{k-1}$ times the reward. Then when a vehicle selects an action $A_t$ to send the BSM, the expected return can be expressed as the following formula:

$$G_t = R_{t+1} + \gamma R_{t+2} + \gamma^2 R_{t+3} + \ldots = \sum_{k=0}^{\infty} \gamma^k R_{t+k+1} \qquad (2)$$

In Equation (2), $G_t$ is the total return and $R_{t+i}$ is the reward at each timestamp, where $i \in \mathbb{N}, i \geq 1$ and $0 \leq \gamma \leq 1$. When $\gamma$ is close to 1, it means we take the future rewards into account more strongly. With RL, vehicles will learn by estimating how good it is to be in a given state or how good it is to perform a given action in a given state in terms of return. The higher the return, the better the state or the action just taken is. The mapping from each state to the probabilities of selecting each possible action is called a policy, denoted as $\pi(a|s)$, which indicates the probability that $A_t = a$ if $S_t = s$. Solving an RL task means, roughly, finding a policy that achieves a high reward value over the long run [13]. At each state, a vehicle can have many actions to choose from, which means it can use different policies to choose an action. The value of taking action $a$ in state $s$ under a policy $\pi$, denoted $q_\pi(s, a)$, is the expected return starting from $s$, taking action $a$, and following policy $\pi$. $q_\pi(s, a)$ is called the state–action value function. The learning target is to find the optimal state–action value function which tells the vehicle the maximum reward it is going to obtain if it is in state $s$ and takes action $a$ under policy $\pi$; it can be denoted as follows:

$$q_*(s, a) = \max_{\pi} q_\pi(s, a) \qquad (3)$$

After finding the optimal policy by $q_*(s, a)$, the vehicle can then pick the action that gives it the optimal state–action value function as follows:

$$\pi_*(a|s) = \begin{cases} 1 & if\ a = \underset{a \in \mathcal{A}}{argmax}\ q_*(s, a) \\ 0 & otherwise \end{cases} \qquad (4)$$

In Equation (4), when the agent is in state $s$, it can just simply select action $a$ which maximizes the value of $q_*(s, a)$ and ignores other optional actions.

### 3.1. Elements of the RL Framework for Congestion and Awareness Control

The primary objective of this paper is to demonstrate the application of RL in selecting appropriate transmission parameters for V2V congestion control. The selection process is modeled as a MDP. This paper will cover the design of the elements of the MDP, considering the following factors:

- Finite state and action space: In the VANET application layer, it is assumed that each vehicle can only choose from a finite set of actions at each state. As the state space is finite, Q-learning can be used to solve the problem.
- Determination of neighboring vehicles: The number of neighboring vehicles is determined based on the interactions with the environment, as indicated by the BSMs received from these vehicles.
- Experimental observations: All observations are calculated through experimental actions taken by the vehicle.
- Independent decision-making: Each vehicle independently selects its actions based on its own observations, without exchanging any information with other vehicles except for BSMs.

The decision-making problem is formalized using the framework of MDP. The following are the key elements of the RL framework used to solve the MDP for V2V congestion control:

- The agent: A learning agent must have the capability to perceive the state of its environment and take actions that can alter it. In the context of this problem, the vehicle acts as the agent that makes the decision of which action to take [13].
- The goal: The objective is to select the optimal action for each state, with the aim of maximizing the reward. A well-defined goal is crucial for the agent (vehicle), such as reducing congestion or enhancing awareness. In this paper, the goal is to maximize the reward of actions that maintain the CBR below 0.6.
- The environment: The environment represents the uncertain world in which the agent operates and interacts. The agent can interact with the environment and modify it through its actions, but it cannot change the rules or dynamics of the environment. In the context of VANETs, the environment encompasses the wireless channel and other vehicles. The uncertainty arises from factors such as dynamic traffic flow, such as changing vehicle velocity and density. The actions of the vehicles can impact the state of the environment (wireless channel status), but will not affect the density of vehicles on the road.
- The action: Actions are the means by which the agent interacts with and influences its environment. In the VANET application layer, the most common actions include setting the transmission power, message transmission rate, or data rate of the messages to be transmitted. In this paper, for simplicity, we only consider the transmission rate. The maximum transmission rate in DSRC is 10 Hz. The action space is defined as 10 discrete transmission rates, ranging from 1 BSM per second to 10 BSMs per second, where $a \in \mathbb{N}, 1 \leq a \leq 10$, for each action $a$.
- The state (observation): The state of the environment in the V2V communication problem is a collection of information that identifies the current situation. This information includes the wireless channel status, such as CBR, BER, IPD, etc. Moreover, the vehicle density, or the number of neighbors of a given vehicle, which represents the dynamics of the environment, should be considered. Even with the same action, the state could be different when the vehicle density is different. In our case, we define the space as a 2-tuple including the CBR and vehicle density, denoted as $s = (CBR, VD)$, $CBR \in \mathbb{R}^+$, $0 \leq CBR \leq 1$, and $VD \in \mathbb{N}$, $1 \leq VD \leq maxVD$. The CBR is a real number between 0 and 1 that represents the channel busy ratio and the vehicle density is the number of vehicles within a 100 m radius, and $maxVD$ is the maximum vehicle density in this range. In this paper, we set $maxVD$ to 50. For each vehicle density, there are 10 CBR values corresponding to the 10 transmission rates. Note that the vehicle density cannot be changed based on the action but only calculated based on the BSMs received from the neighbors, so the whole state space will consist of 500 individual states. At each state, the vehicle selects a new transmission rate from 10 possible rates and updates its state accordingly, based on the information received from its neighbors in the form of BSMs.
- The reward: A reward is a scalar value that measures the quality of an action taken by an agent. The agent uses the rewards provided by the environment after each action to learn and improve its behavior over time [13]. In the context of V2V communication, the reward is calculated based on observations from the environment and the goals of the vehicle. The reward calculation is performed using a reward function, which should be designed to meet the desired learning objectives. The goal of our proposed approach is to maintain the CBR below a predefined threshold $\eta$ while simultaneously maximizing the number of BSMs transmitted. To accomplish this, we have defined the reward function as follows:

$$r(CBR, BR) = BR \cdot CBR \cdot sign(\eta - CBR) \qquad (5)$$

where sign is the signum function shifted by the target value $\eta$. Any action that causes the CBR to exceed $\eta$ will have a negative reward, which can speed up the learning process [13]. A very low transmission rate is not encouraged because the resulting reward will be lower. In this paper, we have used $\eta = 0.6$ as the target channel load.

For different learning objectives, this value can be modified as needed or a different reward function can be implemented.

### 3.2. A Q-Learning Approach

Traditional MDP models require knowledge of state-transition probabilities, which can be challenging to obtain in the context of V2V communication. As an alternative, Q-learning is a model-free algorithm in RL that allows an agent to select and perform actions without relying on a prior understanding of the state-transition probabilities. Instead, the agent learns an optimal policy by directly interacting with the environment. The proposed Q-learning-based congestion control algorithm is implemented in two stages as follows:

- Stage 1: The Q-learning algorithm is implemented using observation data obtained from a simulation. The algorithm generates a Q-table that represents the optimal policies at each state, as demonstrated in Algorithm 1.
- Stage 2: The Q-table generated in the first stage is utilized by the vehicle to determine the interval before the next BSM transmission takes place. This stage is explained in detail in Algorithm 2.

Algorithm 1 provides an overview of our proposed Q-learning-based adaptive congestion control (QBACC) approach. The algorithm starts by initializing all values in the Q-table, which contains all the action–state pairs, to 0. Then at each time step $t$, the vehicle selects an action $a_t$, observes the environment and receives a reward $r_t$, transitions to a new state $s_{t+1}$, and updates the value of $Q(s,a)$ with Equation (6):

$$Q(s_t, a_t) \leftarrow Q(s_t, a_t) + \alpha(r(s_t, a_t) + \gamma \max_a Q(s_{t+1}, a) - Q(s_t, a_t)) \tag{6}$$

In Equation (6), $\alpha$ is the learning rate, where $0 < \alpha \leq 1$, and $\gamma$ is the discount rate, where $0 < \gamma \leq 1$. $Q(s_t, a_t)$ is the current value of $Q(s,a)$, $\max_a Q(s_{t+1}, a)$ is the estimate of optimal future value of $Q(s,a)$ and $r(s_t, a_t)$ is the reward when agent takes action $a$ in state $s$ at time step $t$.

Figure 2 lists the flowchart of Algorithm 1.

---

**Algorithm 1** Q-Learning-based Adaptive Congestion Control (QBACC)

---

**Parameters:** step size $\alpha \in (0,1]$, small $\epsilon > 0$, number of episodes
**Result:** Q-table with values of each state–action pair
 1: Initialize $\mathcal{S}$, the set of states (which contains one state for each beacon rate)
 2: Set $\mathcal{A}(s)$ to be the set of actions that can be taken in state $s$, which consists of the ten possible beacon rates regardless of the value of $s$.
 3: Initialize the Q-table $Q$, where $Q(s,a) = 0$ for all $s \in \mathcal{S}, a \in \mathcal{A}(s)$
 4: **for** each episode **do**
 5:    Set $s$ to be a random state in $\mathcal{S}$
 6:    **for** $i = 1$ to 10 **do**
 7:       Choose action $a$ from $\mathcal{A}(s)$ using the $\epsilon$-greedy algorithm (random chance of choosing the action with the highest value in the Q-table so far; otherwise choose a random action)
 8:       Compute the reward using Equation (5), where CBR is calculated using Equation (7), with the vehicle density as input
 9:       Update $Q(s,a)$ using Equation (6) and the obtained reward
10:       Take action $a$ and move to the corresponding state
11:       Set $s$ to be the new state
12:    **end for**
13: **end for**

---

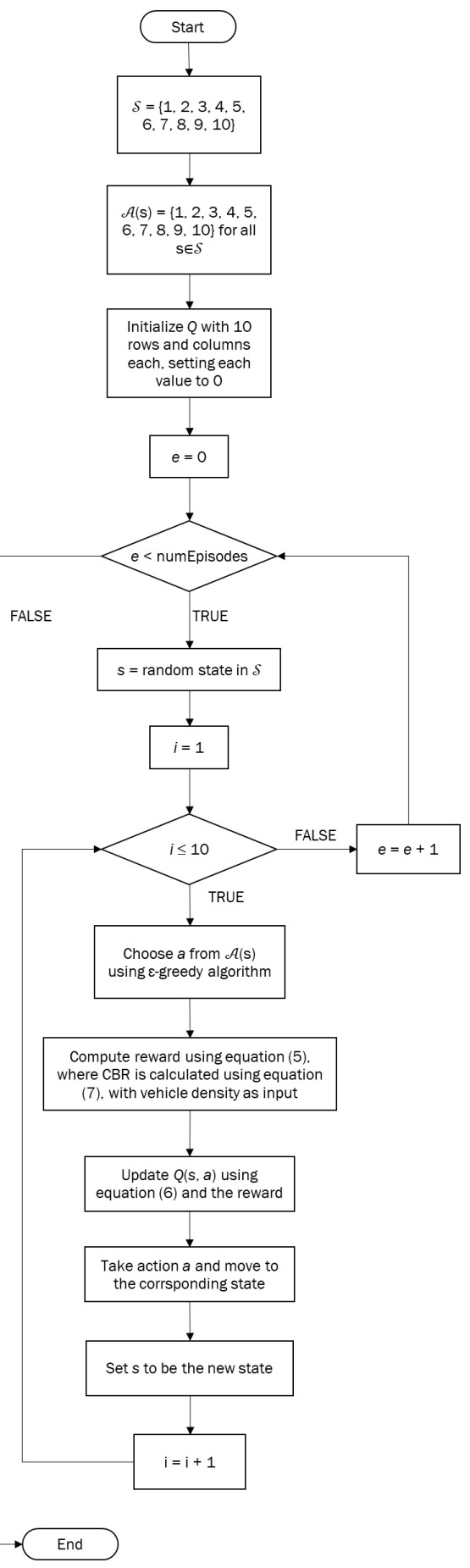

**Figure 2.** Flow Chart of QBACC.

To generate the Q-table, in Algorithm 1, we created 12 different traffic models with different vehicle densities, varying from 0 to $maxVD$ vehicles in a 100 m radius, where we set $maxVD = 50$. Simulations were run for each traffic model with our designed action space, i.e., using a transmission rate ranging from 1 to 10 BSMs per second, to get the observation data (CBR) for each action at each state. These data were then used to generate functions for curves of best fit, which represent the correlation between the average beacon rate used by vehicles and the average CBR experienced by the network. These functions were used to create Equation (7) to estimate the CBR for each transmission rate and density combination based on the known trends. We set a maximum return value of 0.92 to prevent densities greater than the tested range, yielding values greater than 1 since the change in CBR was found to be negligible at high densities. The Q-learning algorithm was then run with the observation data and Equation (7) below to generate the Q-table for each combination of vehicle density and transmission rate for the whole state space. In the following equation, $VD$ is the current vehicle density, $BR$ is the estimated average transmission rate used by surrounding vehicles, and $estCBR$ is the estimated CBR when there are $VD$ neighboring vehicles that use an average transmission rate of $BR$ BSMs.

$$estCBR(VD, BR) = \begin{cases} 0.0101VD + 0.0301 & \text{if } BR = 1 \\ \begin{cases} 0.0189VD + 0.0703 & \text{if } VD \leq 33 \\ 0.2730\ln(VD) - 0.2526 & \text{otherwise} \end{cases} & \text{if } BR = 2 \\ \begin{cases} 0.0249VD + 0.1194 & \text{if } VD \leq 27 \\ 0.1663\ln(VD) + 0.2487 & \text{otherwise} \end{cases} & \text{if } BR = 3 \\ \begin{cases} 0.0314VD + 0.1500 & \text{if } VD \leq 21 \\ 0.0988\ln(VD) + 0.5318 & \text{otherwise} \end{cases} & \text{if } BR = 4 \\ \begin{cases} 0.0379VD + 0.1818 & \text{if } VD \leq 17 \\ 0.0884\ln(VD) + 0.5843 & \text{otherwise} \end{cases} & \text{if } BR = 5 \\ \begin{cases} 0.0686VD + 0.1425 & \text{if } VD \leq 10 \\ 0.0819\ln(VD) + 0.6817 & \text{otherwise} \end{cases} & \text{if } BR = 6 \\ \begin{cases} 0.0772VD + 0.1688 & \text{if } VD \leq 8 \\ 0.0659\ln(VD) + 0.6940 & \text{otherwise} \end{cases} & \text{if } BR = 7 \\ \begin{cases} 0.0843VD + 0.1972 & \text{if } VD \leq 7 \\ 0.0442\ln(VD) + 0.7760 & \text{otherwise} \end{cases} & \text{if } BR = 8 \\ \begin{cases} 0.0891VD + 0.2289 & \text{if } VD \leq 7 \\ 0.0304\ln(VD) + 0.8246 & \text{otherwise} \end{cases} & \text{if } BR = 9 \\ \begin{cases} 0.0930VD + 0.2602 & \text{if } VD \leq 6 \\ 0.0151\ln(VD) + 0.8736 & \text{otherwise} \end{cases} & \text{if } BR = 10 \end{cases} \quad (7)$$

By using this equation to predict the CBR of every combination of vehicle density and transmission rate, the Q-learning algorithm is able to find which combinations yield the best results and can generate the corresponding actions as a result. Each row in the Q-table corresponds to a combination of a vehicle density (ranging from 0 to 50) and an estimated average transmission rate used by the neighboring vehicles (an integer from 1 to 10), while each column represents a transmission rate that can be chosen by the current vehicle.

In Algorithm 1, the first two steps aim to define the state and action space. Step 3 initializes the Q-table and sets each of its values to 0. Since there is no "terminating state", we set the number of episodes to be 80,000 in this paper. During each episode, the vehicle executes steps 4 to 13 to update the Q-table. The algorithm employs a combination of an optimal policy and a random action (with a probability of 0.1) to search for better policies, which is the exploration strategy in RL [13]. The discount factor $\gamma$ and the learning rate $\alpha$ are set to be 0.9 and 0.01, respectively. After 80,000 iterations, the difference between the new and old Q-table is negligible, indicating that the algorithm has converged. The

final Q-table will be saved into a file for use in Stage 2. As noted in [37], the complexity of Algorithm 1 is $O(n)$ in the general case with duplicate actions.

---

**Algorithm 2** Policy Application of QBACC in OMNeT++

---

1: $curCBR$ = Obtain current CBR
2: $curVD$ = Obtain current vehicle density
3: **if** $curVD > maxVD$ **then**
4:     $curVD = maxVD$
5: **end if**
6: $maxVal = -999$
7: $index = 9$ (default value to ensure there is always a valid output)
8: **for** $i = 0$ to 9 **do**
9:     **if** $estCBR(curVD, i + 1)$ using Equation (7) $\geq curCBR$ **then**
10:         $index = i$
11:         **break**
12:     **end if**
13: **end for**
14: **for** $i = 0$ to 9 **do**
15:     $qVal$ = Obtain the entry at index $i$ of the row in the Q-table corresponding to $curVD$ and $index$
16:     **if** $qVal > maxVal$ **then**
17:         $maxVal = qVal$
18:         $bestBeaconRate = i + 1$
19:     **end if**
20: **end for**
21: $bestBeaconInterval = 1/bestBeaconRate$
22: Send beacon using $bestBeaconInterval$

---

Once an optimal policy has been determined in Stage 1, a vehicle can apply this policy to select the BSM transmission rate in Stage 2 (Algorithm 2). In steps 1 and 2, a vehicle detects its environment in terms of CBR and vehicle density. If the measured vehicle density is found to be higher than $maxVD$, it is set to $maxVD$ (steps 3 and 4). In steps 6 and 7, the values of $maxVal$ and $index$ are initialized. $maxVal$ is set to a very small number to ensure that the algorithm will come across a higher value in the Q-table. From steps 8 to 13, the vehicle considers each of the possible average transmission rates from the surrounding vehicles and uses these values with the vehicle density as inputs for Equation (7) to determine which inputs give the smallest estimated CBR value that is greater than or equal to the current CBR. From steps 14 to 20, the vehicle selects the best beacon transmission rate, based on the Q-table given as input. Each row in the Q-table represents a state of the environment with vehicle density, the estimated average transmission rate of surrounding vehicles, and the corresponding estimated CBR, and contains the Q-table values for each possible transmission rate that the vehicle can use. The transmission rate with the maximum value in a row is the best action, i.e., the optimal policy for the corresponding state. An example with selected rows of the Q-table (rather than the entire table) is shown in Table 2; the complete table could not be included due to space limitations.

In steps 14–20, the vehicle checks each possible transmission rate for its current state and selects the one with the highest corresponding value in the Q-table. In step 15, it accesses the row in the Q-table corresponding to the state, and in steps 16-19, it compares it with the maximum Q-table value encountered so far, updating the maximum value if the new value is higher. After it has checked each value in the row, it selects the beacon rate with the highest value. Once this is selected, the beacon interval is calculated, which determines the time before the next BSM is sent and is equal to the reciprocal of the beacon rate. For example, if the current vehicle density is 15, with vehicles sending 1 BSM/s, and the estimated CBR is 0.1816, then using Table 2, the best transmission rate is 3 BSMs/s, with

$qVal = 86$, and the corresponding beacon interval is 0.333 s. Since the iteration number is fixed in Algorithm 2, the time complexity will be just $O(1)$ which will be very quick for the vehicle to select the best action. Figure 3 lists the flow chart of Algorithm 2.

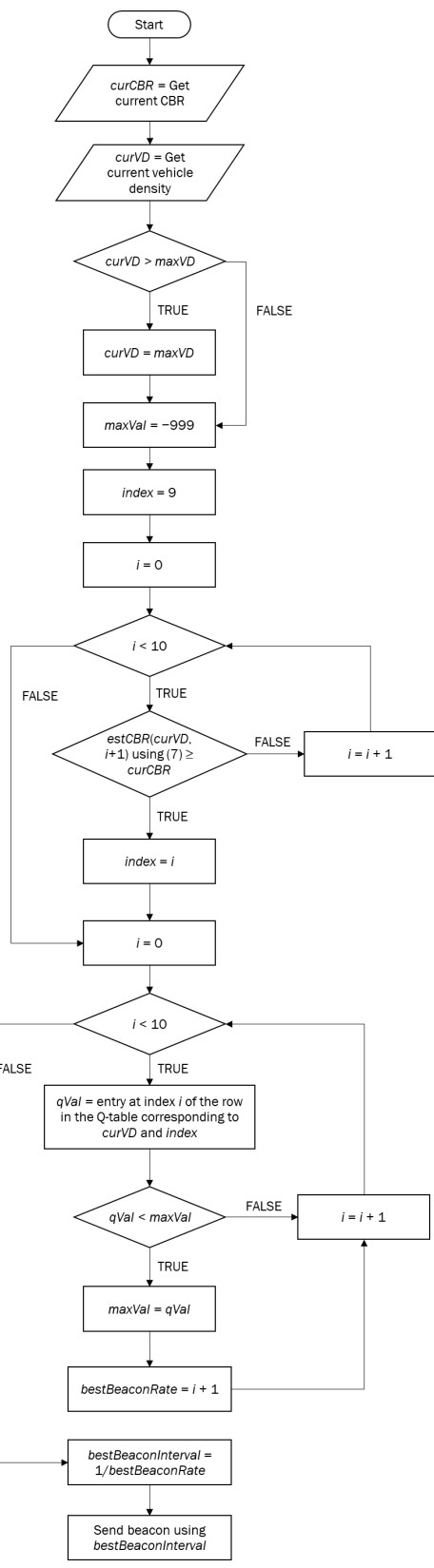

**Figure 3.** Flow Chart of Policy Application of QBACC.

**Table 2.** Example Items of a Q-Table.

| Vehicle Density | Estimated BSM/s | Estimated CBR | Transmission Rate (BSM/s) | | | | |
|---|---|---|---|---|---|---|---|
| | | | 1 | 3 | 5 | 7 | 10 |
| 1 | 1 | 0.0402 | 66 | 71 | 75 | 77 | 82 |
| 1 | 10 | 0.3532 | 68 | 77 | 84 | 89 | 100 |
| 5 | 1 | 0.0806 | 67 | 72 | 77 | 83 | 54 |
| 5 | 10 | 0.7252 | 63 | 60 | 56 | 54 | 13 |
| 15 | 1 | 0.1816 | 71 | 86 | 58 | 61 | 67 |
| 15 | 10 | 0.9145 | 59 | 51 | $-1$ | $-21$ | $-51$ |
| 50 | 1 | 0.5351 | 84 | 71 | 79 | 87 | 100 |
| 50 | 10 | 0.9200 | 73 | 37 | 22 | 8 | $-14$ |

## 4. Evaluation

In this section, we evaluate the performance of the QBACC approach using a framework of Vehicles in Network Simulation (Veins) [38], which contains a basic implementation of the IEEE 802.11p and IEEE 1609 protocols in order to facilitate the testing of V2V networks. Veins connect a widely used network simulation tool, an objective modular network testbed in C++ (OMNeT++) [39], and the traffic mobility simulator simulation of urban mobility (SUMO) [40]. To evaluate the Q-learning performance, we simulated a 20 km highway with four lanes (two in each direction). We only considered data from a 4 km stretch in the middle of the highway to eliminate inaccuracies caused by vehicles entering or exiting the simulation. The simulation involved either 300 or 500 vehicles, with random velocities ranging from 80 to 130 km/h, to produce a dynamic traffic flow with varying vehicle densities. The two vehicle numbers represent low- and high-density traffic scenarios. The simulation time was 1000 s, but we only analyzed data from 350 to 750 s when vehicles were present within the 4 km stretch.

The configuration parameters for the evaluation are summarized in Table 3.

**Table 3.** Configuration Parameters.

| Name | Value |
|---|---|
| Maximum transmission rate | 10 BSM/s |
| Minimum transmission rate | 1 BSM/s |
| Fixed Transmission Power | 20 mW |
| BSM Size | 512 B |
| Data Rate | 6 Mbps |
| Min. Power Level | $-110$ dBm |
| Noise Floor | $-98$ dBm |
| Vehicle Number | 300/500 |
| Highway Length/Lanes | 20 km/4 |
| Simulation Time | 1000 s |

The performance of our proposed QBACC congestion control approach is evaluated by comparing it with other techniques using the dynamic traffic model described above. For all approaches, a fixed transmission power of 20 mW is utilized.

1. 10 Hz: BSMs are transmitted at a constant rate of 10 BSM/s.
2. 5 Hz: BSMs are transmitted at a constant rate of 5 BSM/s.
3. MDPRP: An RL-based congestion control algorithm proposed in [28].

### 4.1. Comparison of CBR

The CBR is defined as the ratio between the time the channel is detected as busy and the total observation time. CBR is a useful indicator of the channel load, with higher values indicating higher channel load and vice versa. To monitor the change in CBR throughout the simulation process, each vehicle calculates the current CBR based on the status of the channel before sending each BSM. Figure 4 illustrates the individual CBR values observed by each vehicle over the simulation interval for the 10 Hz 20 mW scenario. This is referred to as "Real Time CBR", with the X-axis in seconds. It can be difficult to determine overall variations in the channel CBR from this plot. Hence, we introduce a new metric, "average CBR", which calculates the average CBR from all vehicles in 5-second intervals until the end of the simulation time.

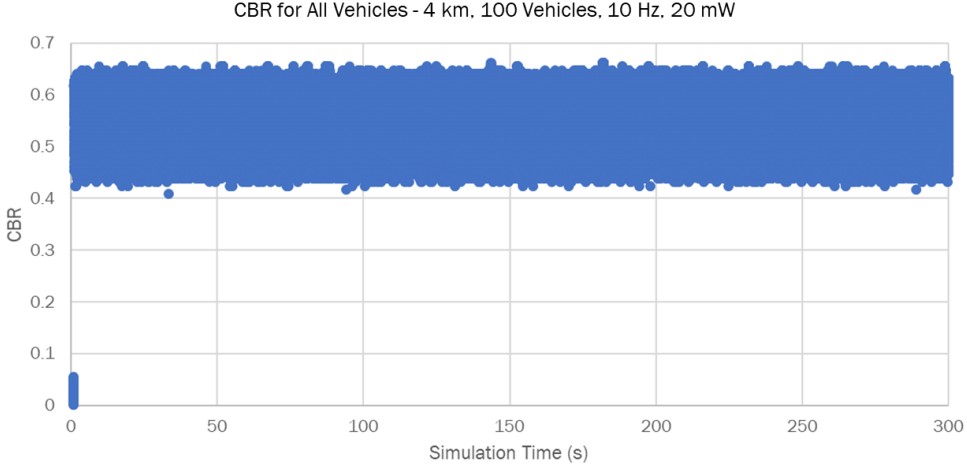

**Figure 4.** Real Time CBR.

Figures 5 and 6 compare the average CBR of the four approaches with 300 and 500 vehicles, respectively. In both cases, the 10 Hz transmission rate consistently results in the highest average CBR because all vehicles use it to send BSMs. The average CBR of 5 Hz is lower than 10 Hz. In both traffic models, QBACC consistently demonstrates stability with all average CBR values remaining below 0.6, which was defined in the reward function. In contrast, MDPRP has a higher average CBR in both traffic models.

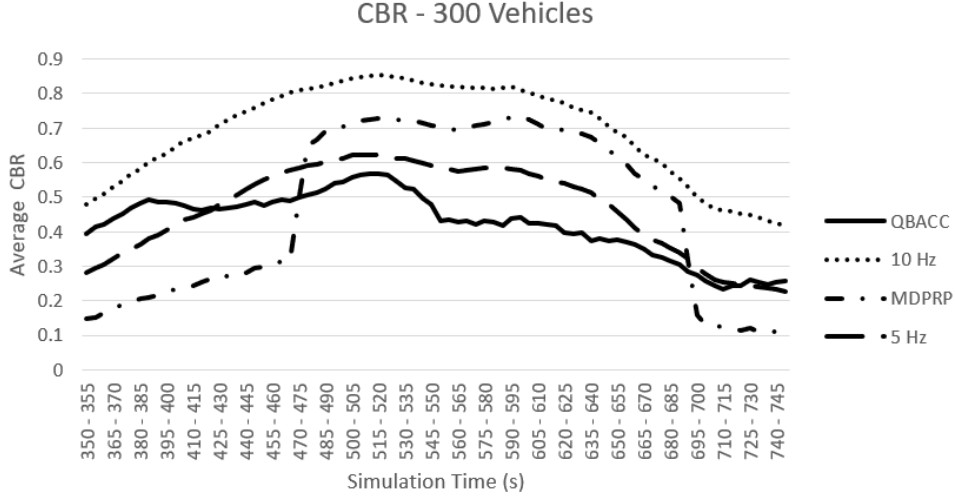

**Figure 5.** Average CBR with 300 Vehicles.

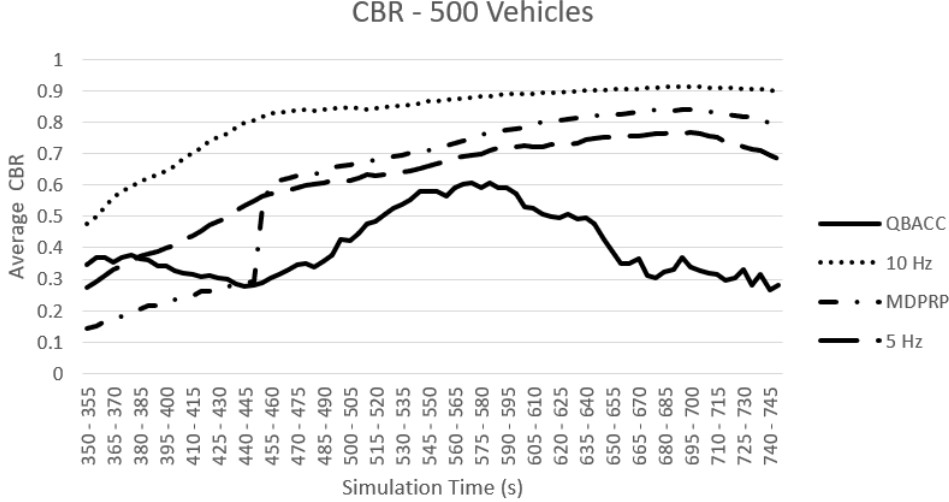

**Figure 6.** Average CBR with 500 vehicles.

### 4.2. Comparison of PDR

In addition to average CBR, we also evaluated other metrics, such as the total number of packets sent, received, and lost during the simulation time. Figures 7 and 8 compare the total number of packets sent by each approach for the low and high vehicle density scenarios, respectively. The PDR represents the percentage of sent packets that were successfully received by a vehicle and is shown in Figures 9 and 10. The proposed QBACC approach has the highest PDR value, at 98.7%, for high vehicle densities, and achieves the second-best results for low vehicle densities, with a PDR of 98.2%, just slightly lower than the PDR of 98.8% obtained using 5 Hz.

The number of lost packets, representing the packets that were not received by any vehicles during the simulation, is shown in Figures 11 and 12. The BER, which calculates the percentage of sent packets that were lost, is shown in Figures 13 and 14. QBACC consistently had low BER values, with the lowest value for 500 vehicles and the second-lowest value for 300 vehicles. The MDPRP BER values were slightly higher than QBACC for both cases; moreover, 5 Hz had slightly lower BER for 300 vehicles, but significantly higher BER for 500 vehicles. A comparison table of total sent packets, total lost packets, BER, and PDR is listed in Table 4.

**Table 4.** Metrics Comparison.

| Metrics | 300 Vehicles | | | | 500 Vehicles | | | |
|---|---|---|---|---|---|---|---|---|
| | **QBACC** | **MDPRP** | **10 Hz** | **5 Hz** | **QBACC** | **MDPRP** | **10 Hz** | **5 Hz** |
| Total sent pkts | 168,807 | 277,769 | 508,930 | 254,465 | 98,272 | 421,124 | 753,100 | 376,550 |
| Total lost pkts | 3061 | 6880 | 27,821 | 2999 | 1293 | 16,011 | 58,379 | 8998 |
| Pkt delivery rate | 0.982 | 0.975 | 0.945 | 0.988 | 0.987 | 0.962 | 0.922 | 0.976 |
| Beacon error rate | 0.018 | 0.025 | 0.055 | 0.012 | 0.013 | 0.038 | 0.078 | 0.239 |

In summary, QBACC has been shown to effectively maintain the channel load below the set threshold of 0.6, as defined in the reward function. This is critical in ensuring the efficient flow of data in high-density vehicular environments. By keeping the channel load below the threshold, QBACC reduces the likelihood of packet losses, resulting in improved network performance, as seen in its BER and Packet PDR performance.



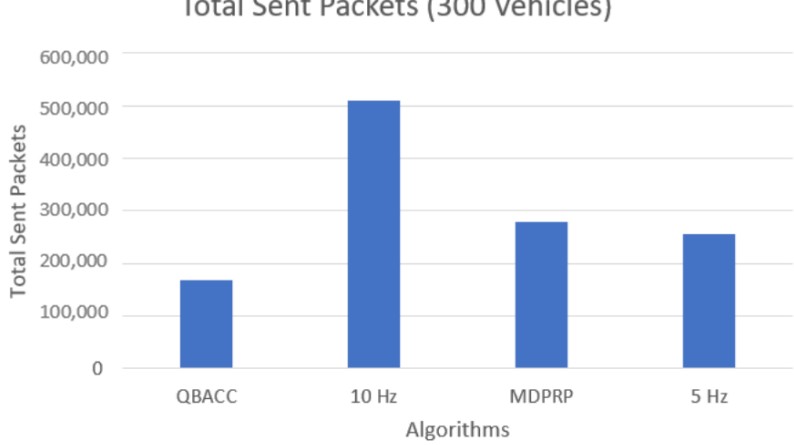

**Figure 7.** Total sent packets with 300 vehicles.

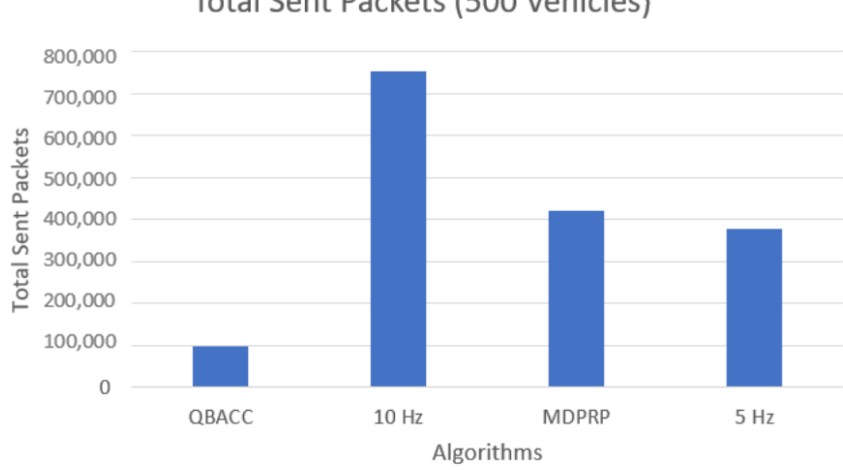

**Figure 8.** Total sent packets with 500 vehicles.

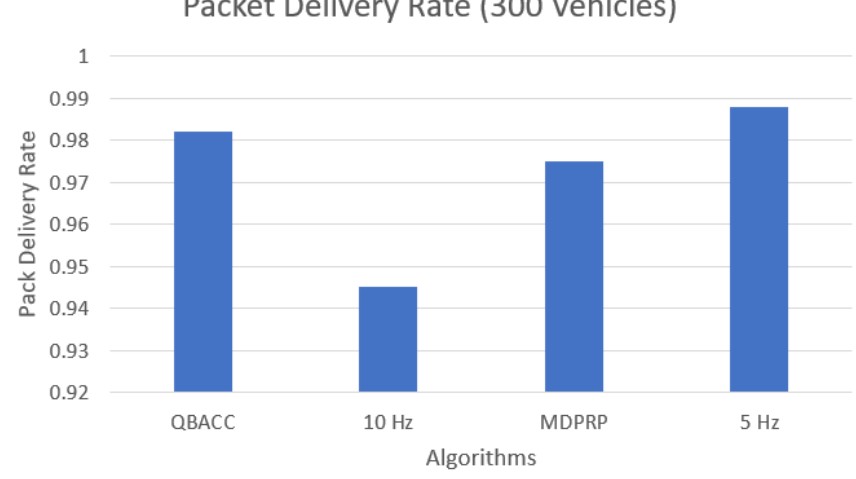

**Figure 9.** Packet delivery rate with 300 vehicles.

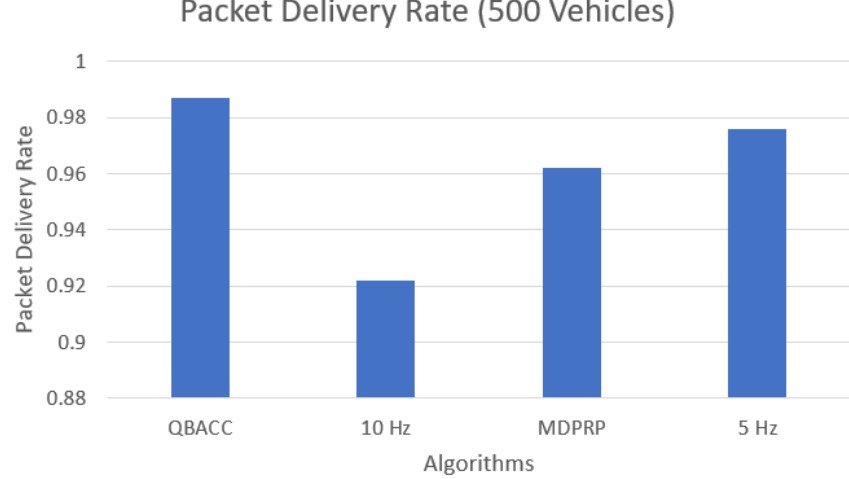

**Figure 10.** Packet delivery rate with 500 vehicles.

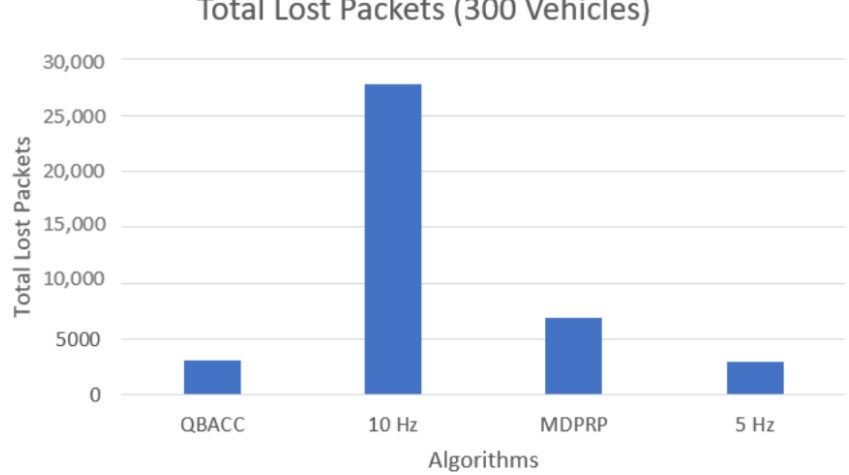

**Figure 11.** Total lost packets with 300 vehicles.

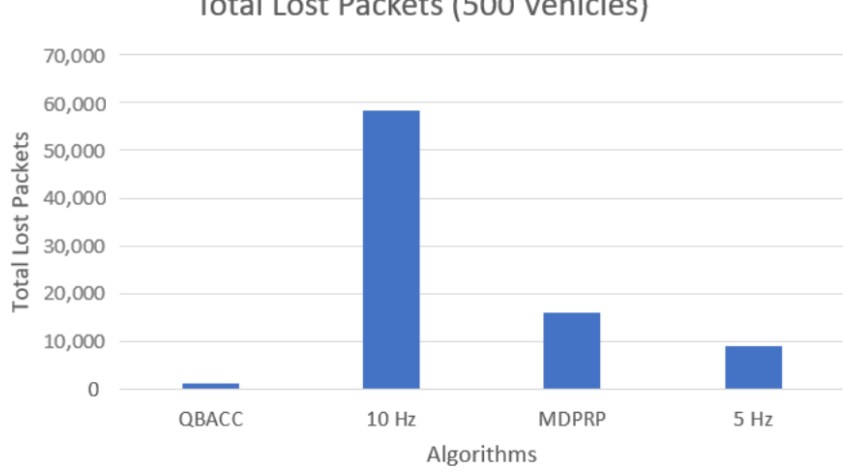

**Figure 12.** Total lost packets with 500 vehicles.

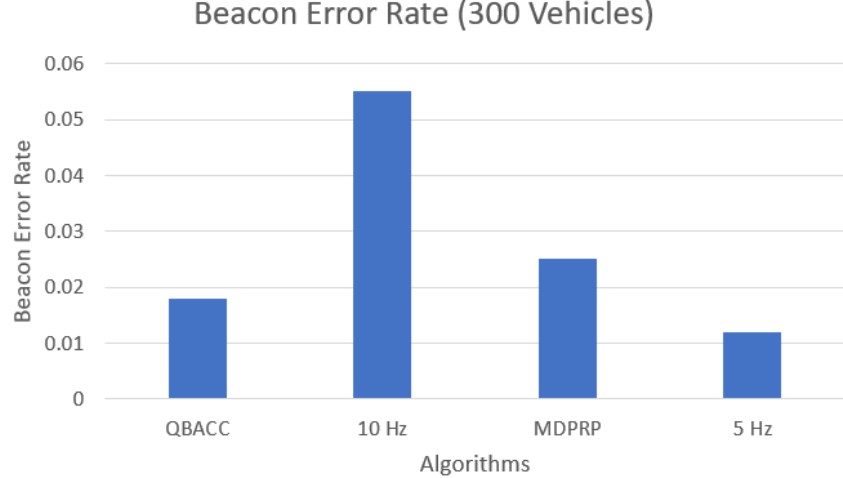

**Figure 13.** Beacon error rate with 300 vehicles.

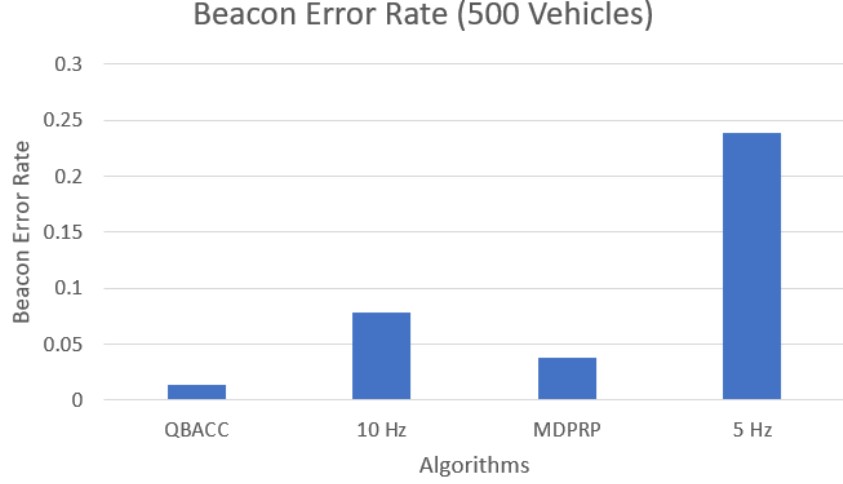

**Figure 14.** Beacon error rate with 500 vehicles.

### 5. Conclusions

The reliable delivery of safety messages in V2V communication requires the maintenance of channel congestion below a critical level. At the same time, an increased transmission rate is necessary for higher awareness. To address this challenge, we present an innovative RL framework that employs Q-learning to determine the most suitable transmission rate policy for BSMs. The aim of our proposed QBACC approach is to strike a balance between maintaining the channel congestion bit rate (CBR) below a specified threshold and maximizing awareness under different traffic conditions. To evaluate QBACC, we used two dynamic traffic models and compared them with existing approaches that employ constant transmission rates and another RL-based approach. The results showed that QBACC outperformed the other approaches by consistently maintaining the channel load at or near the specified level, without exceeding it, for both low and high traffic densities. It also achieved the best performance in terms of packet delivery and beacon error rates for high vehicle densities. For low vehicle density, the constant 5 Hz rate performed the best, but QBACC was a close second, with less than a 1% difference.

We are currently working on enhancing QBACC by designing a comprehensive reward function that takes into account additional metrics, such as inter-packet delay and packet loss. We are also exploring the possibility of adjusting multiple parameters, such as

transmission power and data rate, in addition to the transmission rate, to create a more robust Q-table.

**Author Contributions:** Paper design, X.L. and A.J.; methodology, X.L. and B.S.A.; writing—original draft editing, X.L.; writing—review and editing, A.J., B.S.A. and X.L.; simulation design, X.L.; simulation implementation, B.S.A.; data collection and visualization, X.L. and B.S.A.; data analysis, X.L., B.S.A. and A.J.; Supervision, A.J. funding acquisition, A.J. All authors have read and agreed to the published version of the manuscript.

**Funding:** This research was funded by NSERC DG, grant no. RGPIN-2015-05641.

**Institutional Review Board Statement:** Not applicable.

**Informed Consent Statement:** Not applicable.

**Data Availability Statement:** Not applicable.

**Acknowledgments:** The work of A.J. has been supported by a research grant from the Natural Sciences and Engineering Research Council of Canada (NSERC).

**Conflicts of Interest:** The authors declare no conflict of interest.

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
