# Peer review of "A Reinforcement Learning-Based Congestion Control Approach for V2V Communication in VANET"

_applsci, doi:10.3390/app13063640_

Round 1

Reviewer 1 Report

Authors presented a novel reinforcement learning 10 (RL) based algorithm to solve the MDP and select the most appropriate transmission 11 rate based on the current channel conditions. Paper is very weak in terms of language, technical details and structure. In order to improve the paper, my comments are as follow: 

1. Check spellings and language like line no 11 " alogirithm".

2. Related work should be revised because only authors added two two lines without discussed the limitations of existing work like ". In paper [17], the authors propose 98 a comparative review of congestion control mechanisms. Papers [18,19] introduce theory 99 and methods about general traffic information collection and diffusion from the Internet of 100 Vehicles (IoV) point of view."

3. Add discussion section in related work and add comparison Table in section 2. 

4. All Equations need details like line no 211. 

5. Add flow charts for Algorithm 1 and 2. 

6. The proposed approach should be evaluated with existing solutions. 

7. References are old, add 2022 and 2023 studies. 

8. Paper need improvement in language, technical details and structure as well

9. Add more relevant work in literature like 

DOI: 10.1109/TITS.2020.2994972

https://doi.org/10.3390/app12010476

Author Response

We would like to thank the reviewer for their valuable feedback. We have addressed these comments in the revised manuscript to improve the paper. 

Please check the details in the attachment.

Reviewer 2 Report

Dear Authors

The manuscript entitled "A Reinforcement Learning–Based Congestion Control Approach for V2V Communication in VANET" is presented very well. The overall organization of the manuscript is well defined.

The authors must provide answers to some questions.

There are some critical parameters required while considering V2V communications. 1. vehicle direction; 2. high mobility conditions; 3. vehicle position; Would you consider these parameters for congestion control?

The Markov decision process's reinforcement learning is nicely explained. What are the limitations and drawbacks of existing methodologies? How does reinforcement learning help control congestion in V2V communications? Please explain equation 7 in detail.

Figure 2 depicts "real-time CBR." Kindly mention the units of X-axis CBR.

Kindly mention the comparison table of core parameters required to simulate in OMNET++ before Figure 5. Total packets sent by 300 vehicles Whether 10 Hz and 5 Hz are the types of algorithms The quantitative metrics should be mentioned in the total sent packets description.The number of packets sent is not clear in Figure 5's bar chart. The same remarks apply to figures 6 through 12. Analyze the time complexity of each algorithm and mention it in the conclusion.

Minor revisions are recommended for this manuscript. 

Author Response

(The authors gave the same response as above.)

Round 2

Reviewer 1 Report

Paper is improved in some sections but still not revised as per my previous comments, like 

Point 5: Add flow charts for Algorithm 1 and 2. 

Not added, flow charts should be added in paper

Point 7: References are old, add 2022 and 2023 studies. 

Added 2020 references

Point 9: Add more relevant work in literature like 

Again 2 lines details and added 6 references in a single sentence. 

Some other issues observed in paper like:

1. Authors added 8 references in one line without any technical details "As a result, there has been significant research into congestion 87 control algorithms for BSMs in recent years [15–22]". Line 87 and 88

2. Same issue observed in "To manage conges- 89 tion, rate-based control algorithms aim to reduce this rate to either a fixed predetermined 90 rate [23] or a rate calculated based on the current congestion level [24,25]." Line 89 and 90. 

3. Related work section need to revise from start to end.

Author Response

We would like to thank the reviewers for their valuable feedback. We have addressed these comments in the revised manuscript to improve the paper.

Please check the attachment for more details.

Thank you very much!

Round 3

Reviewer 1 Report

The major weakness of this paper is related work section where the authors only added the summaries of existing work without their limitations and weaknesses. I recommend to revise this section carefully, as this comment is second time. 

Example "The Adaptive Beacon Transmission Power (ABTP) algorithm 136 [23] firstly improves the traditional linear prediction algorithm, and elaborates a linear 137 combination model to reduce the vehicle’s turning error. The approach adjusts transmission 138 power based on vehicle position prediction error, increasing power for vehicles with large 139 errors and reducing it for vehicles with small errors. A comprehensive review of these 140 power control algorithms can be found in [24]." The paper 23 is just introduction whereas the paper 24 is review paper. 

The authors did not revised paper as per my previous comments. 

Author Response

Dear Reviewer, thank you very much for your time. We updated the related work section again according to your comments. Please check the attachment for details.

Best regards
